

# The negativity contour:
# a quasi-local measure of entanglement for mixed states

**Jonah Kudler-Flam[1]⋆, Hassan Shapourian[2,3] and Shinsei Ryu[1]**

**1** Kadanoff Center for Theoretical Physics, University of Chicago, IL 60637, USA
**2** Department of Physics, Harvard University, Cambridge, MA 02138, USA
**3** Department of Physics, Massachusetts Institute of Technology, Cambridge, MA 02139, USA

⋆ jkudlerflam@uchicago.edu

## Abstract

In this paper, we study the entanglement structure of mixed states in quantum many-body systems using the *negativity contour*, a local measure of entanglement that determines which real-space degrees of freedom in a subregion are contributing to the logarithmic negativity and with what magnitude. We construct an explicit contour function for Gaussian states using the fermionic partial-transpose. We generalize this contour function to generic many-body systems using a natural combination of derivatives of the logarithmic negativity. Though the latter negativity contour function is not strictly positive for all quantum systems, it is simple to compute and produces reasonable and interesting results. In particular, it rigorously satisfies the positivity condition for all holographic states and those obeying the quasi-particle picture. We apply this formalism to quantum field theories with a Fermi surface, contrasting the entanglement structure of Fermi liquids and holographic (hyperscale violating) non-Fermi liquids. The analysis of non-Fermi liquids show anomalous temperature dependence of the negativity depending on the dynamical critical exponent. We further compute the negativity contour following a quantum quench and discuss how this may clarify certain aspects of thermalization.

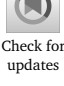

# 1 Introduction

The quantitative study of entanglement has become ubiquitous in the fields of quantum information, condensed matter, and high energy physics. For example, it has played a central role in the characterizations of gapped, critical, topologically-ordered, and holographic quantum systems [1–11]. The most frequent quantification of entanglement in the literature is the von Neumann entropy of the reduced density matrix

$$S(A) = -\text{Tr}\rho_A \log \rho_A, \quad \rho_A = \text{Tr}_{\bar{A}}\rho. \tag{1}$$

It is known that the mutual information ($I(A : \bar{A}) \equiv S(A) + S(\bar{A}) - S(A \cup \bar{A})$) captures all correlation (both classical and quantum) between subregion $A$ and its complement $\bar{A}$. This property makes the von Neumann entropy not particularly useful for characterizing entanglement in mixed states because it fails to distinguish the quantum correlations from classical ones. For instance, when considering systems at finite temperatures, the von Neumann entropy coincides with the standard thermodynamic entropy which is a completely disjoint topic from quantum entanglement. On the other hand, mixed states are omnipresent in many-body quantum systems and it is of great interest to characterize the structure of quantum entanglement in such states.

The logarithmic negativity is a computable candidate measure for the entanglement in mixed states [12–14]. It is motivated from the positive partial transpose (PPT) criterion [15–21] which diagnoses the seperability of mixed states based on if the partially-transposed density matrix is positive semi-definite. The partial transpose of a density matrix with respect to sub-Hilbert space $A$ may be written in terms of its matrix elements in orthonormal basis $\{|e_A^{(k)}\rangle, |e_B^{(j)}\rangle\}$

$$\rho_{AB}^{T_A} = \sum_i \rho_{ijkl} |e_A^{(k)}, e_B^{(j)}\rangle \langle e_A^{(i)}, e_B^{(l)}|. \tag{2}$$

The PPT criterion states that the partial transpose of separable states ($\rho_{AB} = \sum_k p_k \rho_A^k \otimes \rho_B^k$) will be positive semi-definite, while inseparable states will generically have negative eigenvalues. From this observation, one may construct the logarithmic negativity, which measures how negative the eigenvalues are

$$\mathcal{E}_{A;B} = \log \left\| \rho_{AB}^{T_A} \right\|_1, \tag{3}$$

where $\|A\|_1 = \text{Tr}\sqrt{AA^\dagger}$ is the trace norm[1]. We note that for pure states, the logarithmic negativity is equivalent to the Rényi entropy at index $1/2$. Negativity has been used to study a variety of quantum systems including harmonic chains [23–30], spin systems [31–39], 1+1d

---

[1] We focus on the logarithmic negativity instead of its close cousin, *entanglement negativity*, equal to $\frac{e^{\mathcal{E}}-1}{2}$, because logarithmic negativity is generally simpler to work with and interpret as bound on distillable entanglement and a type of entanglement cost [13, 14, 22].

conformal field theories [40–44], 2+1d topologically ordered phases [45–48], variational states [49–52], and holographic field theories [53–56]. Furthermore, the partial-transpose has been exploited to construct topological invariants for symmetry protected topological (SPT) phases protected by anti-unitary symmetries [57–60].

Another disadvantage of the entanglement entropy is that it is a highly non-local object associated to a codimension-1 region of spacetime. In order to have a fine-grained notion for the structure of entanglement in a quantum state, it is desirable to decompose the entanglement entropy into contributions from its real-space degrees of freedom. With this motivation, the authors of Ref. [61] introduced the notion of the *entanglement contour*[2]. The entanglement contour was not originally operationally defined for all quantum systems, but rather defined in terms of a list of heuristic properties that any such entanglement density function should satisfy:

1. Positivity: $s_A(x) \geq 0$.

2. Normalization: $\int_A s_A(x) d^d x = S(A)$.

3. Invariance under spatial symmetries: If T is a symmetry of the reduced density matrix that exchanges positions $x$ and $y$, then $s_A(x) = s_A(y)$.

4. Invariance under local unitaries: If $U_X$ is a local unitary acting only on subsystem $X \subset A$, then the contour $s_A(X)$ associated with density matrix $\rho_A$ is equal to the contour $\tilde{s}_A(X)$ associated with density matrix $U_X \rho_A U_X^\dagger$. Here, $s_A(X)$ is

$$s_A(X) = \int_{x \in X} dx \, s_A(x). \tag{4}$$

5. Upper bound: If $\mathcal{H}_A = \mathcal{H}_\Omega \otimes \mathcal{H}_{\bar{\Omega}}$ and $\mathcal{H}_X \subset \mathcal{H}_\Omega$, then $s_A(X) \leq S(\Omega)$.

By no means do these conditions uniquely specify a contour function $s_A(x)$. In fact, distinct proposals for contour functions for a variety of quantum systems have been studied [61, 63–65]. Motivated by a natural expansion of the entanglement entropy in terms of conditional entropies and interesting fine-grained analysis of the Ryu-Takayanagi surface [66–69], two of us in Ref. [70] proposed a "natural" entanglement contour for generic quantum systems of $n$ (possibly infinite) degrees of freedom $\{A_i\}$ with generic entangling surface geometries

$$s_A(A_i) = \frac{1}{2} \left[ S(A_i | A_1 \cup \cdots \cup A_{i-1}) + S(A_i | A_{i+1} \cup \cdots \cup A_n) \right], \tag{5}$$

where $S(A|B)$ is the conditional entropy

$$S(A|B) \equiv S(A \cup B) - S(B). \tag{6}$$

This admits a differential form for continuous systems [70]

$$s_A(x) = \frac{1}{2} \left( \frac{\partial S(x_1, x)}{\partial x} - \frac{\partial S(x, x_2)}{\partial x} \right), \tag{7}$$

where $x_1$ and $x_2$ are the positions of the endpoints $A$. This was generalized to spherical regions in higher dimensions in Ref. [69]. Remarkably, in the scaling limit, this computable definition of the entanglement contour coincides with the more complex contour functions studied in

---

[2]We note that a similar construction for the harmonic chain was studied earlier in Ref. [62].

Refs. [61, 63–65][3]. For a single interval of length $l$ in 1+1d conformal field theories at finite temperature, the entanglement contour is found to be

$$s_A(x) = \frac{\pi c}{6\beta} \left( \coth\left( \frac{\pi(x + \frac{l}{2})}{\beta} \right) + \coth\left( \frac{\pi(\frac{l}{2} - x)}{\beta} \right) \right). \tag{8}$$

This manifestly displays the extensivity of the von Neumann entropy at high temperatures because the contour is approximately constant and finite away from the entangling surface. Naturally, one would like to compute a contour function that only captures the quantum correlations to understand the quasi-local structure of entanglement. We therefore consider a contour for negativity in this paper. This was initially considered in Ref. [71] for free bosonic lattice systems though technical difficulties arose regarding non-positivity of the candidate contour function. It was further considered holographically using bit threads in Ref. [70] in a way that satisfied the axioms of the contour though is technically quite difficult to compute for generic configurations. It is then of great interest to demonstrate a *computable* negativity contour that works for general quantum systems and subsystem configurations analogous to the entanglement contour function (5).

Among various quantum systems, the free Fermi gas contains rich patterns of entanglement despite its simplicity. While most systems obey an area-law entanglement [72] (especially in higher dimensions), Fermi gases are an exception and violate the area-law in all dimensions thanks to their codimension-one Fermi surface [73–80]. Furthermore, it is tempting to attribute the stability of Fermi gases against weak interactions to this pattern of long-range entanglement [81, 82]. At the same time, Fermi gases are relatively easy to analyze exactly and hence provide a good platform to benchmark our proposals for the negativity contour.

It is believed that Fermi liquids are described by similar patterns of entanglement as well [81–84]. On the other hand, more exotic compressible states such as non-Fermi liquids also logarithmically violate the area-law entanglement, although they are fundamentally different and are described by emergent Fermi surfaces coupled to gauge fields [85–91]. In this paper, we study a class of holographic non-Fermi liquids [92–99] and show that the negativity contour uncovers certain differences between the Fermi gases and non-Fermi liquids.

In addition, the negativity contour provides an elucidating probe of thermalization, namely the crossover from quantum to classical behavior of subregions. Understanding the mechanism of thermalization is a challenge of great interest to both statistical and high-energy physicists. The former community is concerned with the emergence of statistical mechanics and hydrodynamics from isolated quantum systems. On the other hand, thermal states in conformal field theories are conjectured to be dual to black hole states in asymptotically *AdS* spacetimes [100–102]. Presumably, one can then learn about black holes and quantum gravity by studying thermalization in their non-gravitational quantum field theory counterparts. To probe thermalization, we study global quantum quenches i.e. a sudden change of the Hamiltonian from gapped to gapless. While the signature of thermalization in the entanglement contour is a constant (8), we find the signature in the negativity contour to be a relaxation to zero away from the entangling surface (when restricting to sufficiently small subsystems). This makes sense physically because after thermalization, subsystems should appear classical i.e. they should have vanishing entanglement with their surroundings. While we compute the negativity only for free fermions (a trivially integrable system), we argue that the negativity contour will be illuminating for non-integrable systems where the dynamics of quantum information are highly non-trivial.

The rest of the paper is organized as follows: In section 2, we present our proposal for a computable negativity contour. In section 3, we study systems with a Fermi surface and branch

---

[3]We do not have a first principles proof of this, but have numerical evidence for this claim displayed in Appendix B.

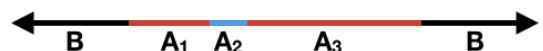

Figure 1: Configuration of subsystems in Eq. (9).

out to novel non-Fermi liquids in section 4. In Section 5, we introduce dynamics and analyze our numerical results using the quasi-particle picture. Finally, we discuss our results and avenues for future work in Section 6. We present details of our calculations in the appendices.

## 2 Negativity contour

We are able to construct a negativity contour for free fermions that satisfies the axioms of a contour function. Our approach draws on the original entanglement contour approach for free fermions from Ref. [61] where the contour function is defined by a weighting of the eigenvalues of the reduced density matrix according to the magnitudes of the single-particle eigenfunctions. The details of our negativity contour for Gaussian states are explained in Appendix A.

We propose a definition of the negativity contour for generic many-body quantum systems that is straightforward to compute[4]

$$e_A(A_2) \equiv \frac{1}{2}\left(\mathcal{E}_{A_1 \cup A_2; B \cup A_3} - \mathcal{E}_{A_3; B \cup A_1 \cup A_2} + \mathcal{E}_{A_2 \cup A_3; B \cup A_1} - \mathcal{E}_{A_1; B \cup A_2 \cup A_3}\right) \tag{9}$$

and admits the differential form if we restrict to continuum field theories[5]

$$e_A(x) = \frac{1}{2}\left(\frac{\partial \mathcal{E}(x_1, x)}{\partial x} - \frac{\partial \mathcal{E}(x, x_2)}{\partial x}\right), \tag{10}$$

where $x_1$ and $x_2$ denote the left and right end points of $A$, respectively, and $\mathcal{E}(x, y)$ corresponds to the logarithmic negativity between subregion $(x, y)$ and its complement (see Fig. 1). In analogy to (7), this may be generalized to highly symmetric regions in higher dimensions in the spirit of Ref. [69]. Because logarithmic negativity does not satisfy strong subadditivity, this contour function is not strictly positive. Even so, we find that it is a faithful approximation to a negativity contour and furthermore appears to match with the Gaussian state contour in the scaling limit. An additional simple sanity check is to consider $A$ as the entire system. Whether or not $A$ is in a mixed state, the above contour will be zero everywhere. This is because when $B = \emptyset$, eqn. (9) reduces to

$$e_A(A_2) = \frac{1}{2}\left(\mathcal{E}_{A_1 \cup A_2; A_3} - \mathcal{E}_{A_3; A_1 \cup A_2} + \mathcal{E}_{A_2 \cup A_3; A_1} - \mathcal{E}_{A_1; A_2 \cup A_3}\right), \tag{11}$$

which is manifestly zero because $\mathcal{E}_{A;B} = \mathcal{E}_{B;A}$.

Let us work out the negativity contour in a 1+1d CFT at finite temperature to understand its general behavior and distinctions from the entanglement contour. The logarithmic negativity for a single interval $(-l/2, l/2)$ at finite inverse temperature $\beta$ is computed using the following four-point function [42]

$$\mathcal{E} = \lim_{L \to \infty} \lim_{n_e \to 1} \log \langle \sigma_{n_e}(-L) \bar{\sigma}_{n_e}^2(-l/2) \sigma_{n_e}^2(l/2) \bar{\sigma}_{n_e}(L) \rangle_\beta, \tag{12}$$

---

[4]We can similarly consider a "contour" function for the mutual information constructed from linear combinations of the entanglement contour proposed in [70]. We use quotation marks because this contour does not strictly satisfy the positivity condition. For subsystems $A$ and $B$, the mutual information "contour" is defined as $mi_A(x) \equiv s_A(x) - s_{A \cup B}(x)$.

[5]One must be careful with the derivatives of the logarithmic negativity due to its non-analyticity derived from the use of the trace norm. See Ref. [103] for important discussion on taking derivatives of the negativity (in particular the trace norm).

where the twist-fields are conformal primaries of dimensions

$$h_{\sigma_{n_e}} = h_{\bar{\sigma}_{n_e}} = \frac{c}{24}\left(n_e - \frac{1}{n_e}\right), \quad h_{\sigma^2_{n_e}} = h_{\bar{\sigma}^2_{n_e}} = \frac{c}{24}\left(\frac{n_e}{2} - \frac{2}{n_e}\right).$$ (13)

The final result for a general CFT is given by [42]

$$\mathcal{E} = \frac{c}{2}\log\left[\frac{\beta}{\pi\epsilon}\sinh\left(\frac{\pi l}{\beta}\right)\right] - \frac{\pi cl}{2\beta} + f\left(e^{-2\pi l/\beta}\right) + 2\log c_{1/2},$$ (14)

where we have introduced the regulator $\epsilon$ and constant $c_{1/2}$. The function $f(x)$ is theory-dependent because we are working with a four-point function that depends on full operator content. In general, $f(x)$ is unknown, though it tends to zero when $l \ll \beta$ and to a constant when $l \gg \beta$. Dropping the non-universal term, the negativity contour is found to be

$$e_{univ}(x) = \frac{\pi c}{4\beta}\left(\coth\left(\frac{\pi(x + \frac{l}{2})}{\beta}\right) + \coth\left(\frac{\pi(\frac{l}{2} - x)}{\beta}\right) - 2\right).$$ (15)

This should be compared to the entanglement contour (8) which is proportional except for the last term in the parentheses. This term effectively cancels the thermal contribution to the entropy so that the negativity contour exponentially decays to zero at distances larger than the inverse temperature from entangling surface. Thus, the inverse temperature acts as the relevant length scale for quantum correlations to remain when heating up the system[6].

In the special case of holographic conformal field theories, the negativity including the function $f(x)$ can be solved using the approximation that logarithmic negativity is proportional to the entanglement wedge cross-section in holographic theories

$$\mathcal{E} = \mathcal{X}^d_{hol}\frac{E_W}{4G_N},$$ (16)

where $\mathcal{X}^d_{hol}$ is dependent on the geometry of the entangling surface. For more details, see the derivation of the holographic formula for $AdS_3/CFT_2$ in Ref. [105]. Using (16), one finds that

$$\mathcal{E} = \frac{c}{2}\min\left[\log\left(\frac{\beta}{\pi\epsilon}\sinh\frac{\pi l}{\beta}\right), \log\left(\frac{\beta}{\pi\epsilon}\right)\right].$$ (17)

The transition occurs $l/\beta = \log(1 + \sqrt{2})/\pi \simeq 0.28$ . Using (10), we find

$$e(x) = \begin{cases} \frac{\pi c}{4\beta}\left[\coth\left(\frac{\pi(x+\frac{l}{2})}{\beta}\right) + \coth\left(\frac{\pi(\frac{l}{2}-x)}{\beta}\right)\right], & \frac{l}{2} - |x| < \frac{\beta\log(1+\sqrt{2})}{\pi} \\ 0, & \frac{l}{2} - |x| > \frac{\beta\log(1+\sqrt{2})}{\pi} \end{cases}.$$ (18)

Though quite similar to the universal part of the negativity contour, we find a sharp drop-off at $l \sim \beta$ rather than a smooth exponential decay.

## 3 Fermi surface systems

In this section, we study the negativity contour for finite-temperature states of free fermions. The Rényi entropies of a subregion of characteristic size $l$ for a $(d + 1)$ metallic system with a codimension-one Fermi surface is [74],

$$S_n(l) = C_d\left(\frac{n+1}{6n}\right)l^{d-1}\log l,$$ (19)

---

[6]We refer the reader to a related "quantum correlation length" that has recently been studied using tripartite logarithmic negativity in Ref. [104]. Both the negativity contour and this correlation length scale linearly with $\beta$ in critical systems. It is of interest to elucidate the connection between these quantities.

where

$$C_d = \frac{1}{4(2\pi)^{d-1}} \int_{\partial\Omega} \int_{\partial\Gamma} dS_k dS_x |n_x \cdot n_k|, \tag{20}$$

$\Omega$ is the volume of the subregion normalized to one, $\Gamma$ is the volume enclosed by the Fermi surface, and the integration is carried out over the surface of both domains.

In particular, the entanglement entropy of a cylinderical region of perimeter $L$ and length $l$ in a $(2+1)$ metal reads as

$$S_n(l) = \left(\frac{n+1}{6n}\right) C_2 \cdot L \log l. \tag{21}$$

The filled Fermi surface of a $(2+1)$ metal may be viewed as a collection of $(1+1)$ gapless modes [78, 106–108] and the entanglement can be understood as a sum of one-dimensional segments ($L$ of them) each of which contributes $(n+1)/6n \cdot \log(l)$ up to a geometrical coefficient (20). The above formula was shown to be in a remarkable agreement with numerical simulations of various microscopic lattice models [75, 76]. The finite temperature Rényi entropy in $(1+1)$ free fermions has the same form as the zero temperature entropy provided that we replace $\log(l)$ by $\log[(\beta/\pi)\sinh(\pi l/\beta)]$. We can further follow similar lines of argument to deduce that the Rényi entropy of a $(2+1)$ metal should obey the following form [78, 82],

$$S_n(l, T) = \left(\frac{n+1}{6n}\right) C_2 \cdot L \log \left| \frac{\beta}{\pi\epsilon} \sinh\left(\frac{\pi l}{\beta}\right) \right|. \tag{22}$$

As shown in Ref. [109], the negativity of codimension-1 free fermions obey a similar form in terms of the one dimensional negativity. So, for finite temperature negativity we can write

$$\mathcal{E}(l, T) = C_d \cdot \frac{l^{d-1}}{2} \left[ \log\left(\frac{\beta}{\pi\epsilon} \sinh\left(\frac{\pi l}{\beta}\right)\right) - \frac{\pi l}{\beta} \right]. \tag{23}$$

We should note that the bipartite logarithmic negativity is equal to the 1/2-Rényi entropy at zero temperature. However, there is an important difference between the logarithmic negativity and the Rényi entropy. Entanglement negativity has an extra term linear in $l$ inside the parenthesis compared to the Rényi entropies and this term exactly cancels the volume law term in the high temperature limit, i.e., the entanglement negativity obeys an area law $\mathcal{E}(lT \gg 1) \propto l^{d-1} \log(\beta/\pi\epsilon)$ while Rényi entropy grows as a volume law.

Plugging Eq. (23) into the derivative formula (10), the negativity contour is found to be

$$e_A(x) = C_d \cdot \frac{\pi l^{d-1}}{4\beta} \left[ \coth\left(\frac{\pi(x+\frac{l}{2})}{\beta}\right) + \coth\left(\frac{\pi(\frac{l}{2}-x)}{\beta}\right) - 2 \right]. \tag{24}$$

We compare the Gaussian formula (as explained in Appendix A) and the discrete derivative formula on the lattice (9) with the above continuum limit expression for a free fermion chain in Fig. 2 where we observe a good agreement. It is worth looking at two extreme behaviors of the scaling limit formula which are also evident in Fig. 2. First, near entangling boundaries in the regime where $l/2 - |x| \ll \beta$, the negativity contour admits a power law

$$e_A(x) \to C_d \cdot l^{d-1} \frac{l/4}{l^2/4 - x^2}, \tag{25}$$

which appears as a linear behavior in Fig. 2 for large values of $\frac{l/2}{l^2/4-x^2}$ (horizontal axis of Fig. 2). Second, far from the entangling boundaries in the regime where $l/2 - |x| \gg \beta$, the negativity contour obeys the form

$$e_A(x) \to C_d \frac{\pi l^{d-1}}{\beta} \exp\left(-\frac{\pi l}{2\beta}\right) \cosh\left(\frac{\pi x}{\beta}\right), \tag{26}$$

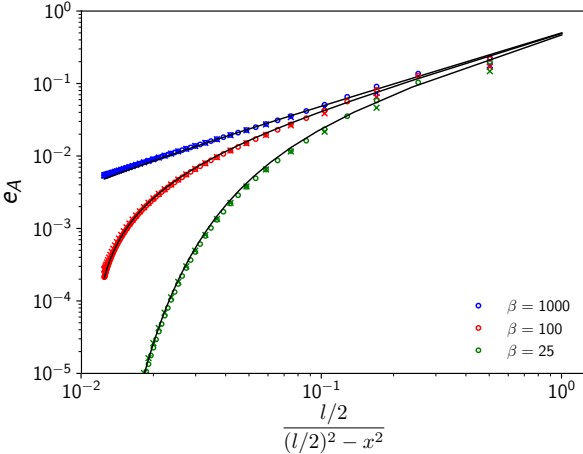

Figure 2: Comparison of Gaussian formula (circles), continuum (solid curves) and lattice derivative (crosses) formulae for negativity contour of a single interval (of length $l$) in a free fermion chain at temperature $T = 1/\beta$. Here, total system is $L = 400$ and subsystem size is $l = 160$. The continuum formula is given in (15).

which is exponentially small in $l/\beta$. In the next section, we study a class of holographic non-Fermi liquids where we find that the negativity contour in the latter region (which is exponentially small for the free Fermi gas) vanishes (possibly up to $1/N$ corrections in the holographic calculations).

## 4 Non-Fermi liquids

To study Fermi-surface systems holographically, we follow [92] in making use of general hyperscaling violating quantum field theories with the following scaling of the coordinates[7]

$$x_i \to \lambda x_i, \quad t \to \lambda^z t, \quad ds \to \lambda^{\theta/d} ds, \tag{27}$$

where $d$ is the number of spatial dimensions. These theories have gravity duals with metrics of the form

$$ds^2 = \frac{R^2}{r^2}\left(-f(r)dt^2 + g(r)dr^2 + dx_i^2\right), \tag{28}$$

where $R$ is the $AdS$ radius and

$$f(r) = f_0 r^{-2d(z-1)/(d-\theta)}, \quad g(r) = g_0 r^{2\theta/(d-\theta)}, \tag{29}$$

when the field theory is at zero temperature. We will not be concerned with $f_0$ because we will always be working on a constant time slice, but the value of $g_0$ depends on the dimension and hyperscaling violating parameters

$$g_0 = (z-1)^{-\frac{\theta}{d(d-\theta)}}(z+d-\theta-1)^{1+\frac{\theta}{d(d-\theta)}}(z+d-\theta)\frac{d^2}{(d-\theta)^2}. \tag{30}$$

---

[7]We note that negativity has been computed in hyperscaling violating harmonic models in Ref. [110].

Furthermore, this family of metrics admit black hole solutions describing the boundary field theory at finite temperature when we replace $f(r)$ and $g(r)$ by

$$f(r) = f_0 r^{-2d(z-1)/(d-\theta)} \left[ 1 - \left( \frac{r}{r_H} \right)^{d(1+z/(d-\theta))} \right], \tag{31}$$

$$g(r) = g_0 r^{2\theta/(d-\theta)} \left[ 1 - \left( \frac{r}{r_H} \right)^{d(1+z/(d-\theta))} \right]^{-1}, \tag{32}$$

where $r_H$ is the value of the radial coordinate at the black hole horizon. We consider a strip geometry of the boundary theory

$$x_1 \in \left[ -\frac{l}{2}, \frac{l}{2} \right], \quad x_{i \neq 1} \in \left[ -\frac{L}{2}, \frac{L}{2} \right], \quad L \gg l. \tag{33}$$

We denote the total cross-sectional area of the strip as $\Sigma$ and set

$$\theta = d - 1, \tag{34}$$

which is a constraint found for holographic theories of compressible states with hidden Fermi surfaces [92].

In the high-temperature limit, the holographic negativity is simple to compute because the entanglement wedge cross section is purely radial, terminating on the black hole horizon

$$E_W = 2\Sigma R g_0^{1/2} \times \int_0^{r_H} dr \frac{r^{d-2}}{\sqrt{1 - \left( \frac{r}{r_H} \right)^{d(1+z)}}} \tag{35}$$

$$= \begin{cases} \dfrac{4\Sigma R g_0^{1/2} \tanh^{-1} \left[ \sqrt{1 - \left( \frac{\epsilon}{r_H} \right)^{1+z}} \right]}{1+z} & d = 1 \\[4ex] \dfrac{2\Sigma R g_0^{1/2} \sqrt{\pi} \Gamma \left[ 1 + \frac{d-1}{d(1+z)} \right]}{(d-1)\Gamma \left[ \frac{1}{2} + \frac{d-1}{d(1+z)} \right]} r_H^{d-1} & d > 1 \end{cases}, \tag{36}$$

where we have introduced UV regulator $\epsilon$. The horizon radius and the temperature are related as $r_H^d \sim T^{-1/z}$, so we find an anomalous power-law scaling of the logarithmic negativity at high temperatures

$$\mathcal{E} = \begin{cases} \dfrac{c\Sigma R g_0^{1/2} \tanh^{-1} \left[ \sqrt{1 - T^{\frac{1+z}{z}} \epsilon^{1+z}} \right]}{1+z} & d = 1 \\[4ex] \dfrac{c\Sigma R g_0^{1/2} \sqrt{\pi} \Gamma \left[ 1 + \frac{d-1}{d(1+z)} \right]}{2(d-1)\Gamma \left[ \frac{1}{2} + \frac{d-1}{d(1+z)} \right]} T^{\frac{1-d}{dz}} & d > 1 \end{cases}, \tag{37}$$

where $c \equiv 3R/2G_N$ is the central charge when $d = 1$, but related to the trace anomaly in higher dimensions. Similar temperature scaling was also found in the capacity of entanglement [111]. At low temperatures, the holographic negativity is proportional to the holographic entanglement entropy studied in [92]

$$S_{lT^{1/z} \to 0} \sim cQ^{(d-1)/d} \Sigma \log(Q^{1/d} l), \tag{38}$$

where $Q$ is the total charge density which is related to the Fermi wavevector as $Q \sim k_F^d$. For reference, the high temperature result was

$$S_{lT^{1/z} \to \infty} \sim cQ^{(d-1)/d} \Sigma l T^{1/z}. \tag{39}$$

With these results, we can quantitatively describe the form of the entanglement contour

$$
s_A(x) \sim \begin{cases} cQ^{(d-1)/d} \frac{l}{\frac{l^2}{4}-x^2}, & \frac{l}{2}-|x| \lesssim T^{-1/z} \\ cQ^{(d-1)/d} T^{1/z}, & \frac{l}{2}-|x| \gtrsim T^{-1/z} \end{cases}.
\tag{40}
$$

Equation (9) leads to

$$
e_A(x) \sim \begin{cases} cQ^{(d-1)/d} \frac{l}{\frac{l^2}{4}-x^2}, & \frac{l}{2}-|x| \lesssim T^{-1/z} \\ 0, & \frac{l}{2}-|x| \gtrsim T^{-1/z} \end{cases}.
\tag{41}
$$

Similar to what we found for free fermions, the negativity contour behaves as the entanglement contour except that it has a sharp cutoff at a distance related to the temperature away from the entangling surface[8]. There must also be some intermediate regime of the negativity contour in holographic systems in order to reproduce the total result for negativity when integrating the contour. Unfortunately, due to the complexity of the bulk metric, we are unable to resolve this interesting intermediate regime analytically. Understanding the length scale at which quantum correlations disappear at finite temperature for generic hyperscaling violating theories is thus an open question.

The mutual information "contour" is approximately

$$
mi_A(x) \sim \begin{cases} cQ^{(d-1)/d} \left( \frac{l}{\frac{l^2}{4}-x^2} - T^{1/z} \right), & \frac{l}{2}-|x| \lesssim T^{-1/z} \\ 0, & \frac{l}{2}-|x| \gtrsim T^{-1/z} \end{cases}.
\tag{42}
$$

This is proportional to the negativity contour except for the temperature dependent term near the entangling surface.

# 5 Quantum quench and thermalization

The entanglement structure of non-equilibrium states is significantly less well understood than that of ground states and thermal states, though it has recently peaked great interest in the context of thermalization, quantum information scrambling, and black holes. Elucidating the mechanism for the thermalization of many-body quantum systems is important for understanding the emergence of classical thermal physics from a pure quantum state. Practically, classical physics is tractable to simulate using classical computation while highly entangled quantum phenomena are notoriously difficult to simulate. We are interested in how we can treat subsystems classically after quantum thermalization. It turns out that logarithmic negativity is the relevant entanglement measure for this purpose, rather than von Neumann entropy, as it captures only the quantum entanglement, discarding the innocuous classical correlations. We study the negativity contour following a simple mass quench of free fermions[9]. We partition our system into two finite subsystems (denoted by A and B in Fig. 3) and treat the rest of the system (denoted by C in Fig. 3) as a thermal bath.

We prepare our system in the ground state of the gapped Su-Schriffer-Heeger (SSH) Hamiltonian

$$
\hat{H}_{SSH} = -\sum_i \left[ t_2 f_{i+1}^{L\dagger} f_i^R + t_1 f_i^{L\dagger} f_i^R + \text{h.c.} \right],
\tag{43}
$$

---

[8]Such sharp transitions are ubiquitous in (large $N$) holographic theories and are expected to be smoothed out by $1/N$ corrections.

[9]We note that the entanglement contour has also been analyzed in momentum space following certain quenches where it has been observed that the entanglement entropy is dominated by modes closest to the Fermi surface [112].

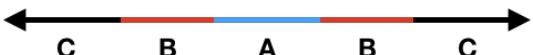

Figure 3: We partition our system into three regions. After the global quantum quench, $C$ effectively acts as a thermal bath for $A \cup B$. While the von Neumann entropy of region $A$ or $B$ satisfies a volume law, the entanglement between the two regions (characterized by $\mathcal{E}_{A:B}$) is negligible. Hence, system $A \cup B$ approaches a "classical" state.

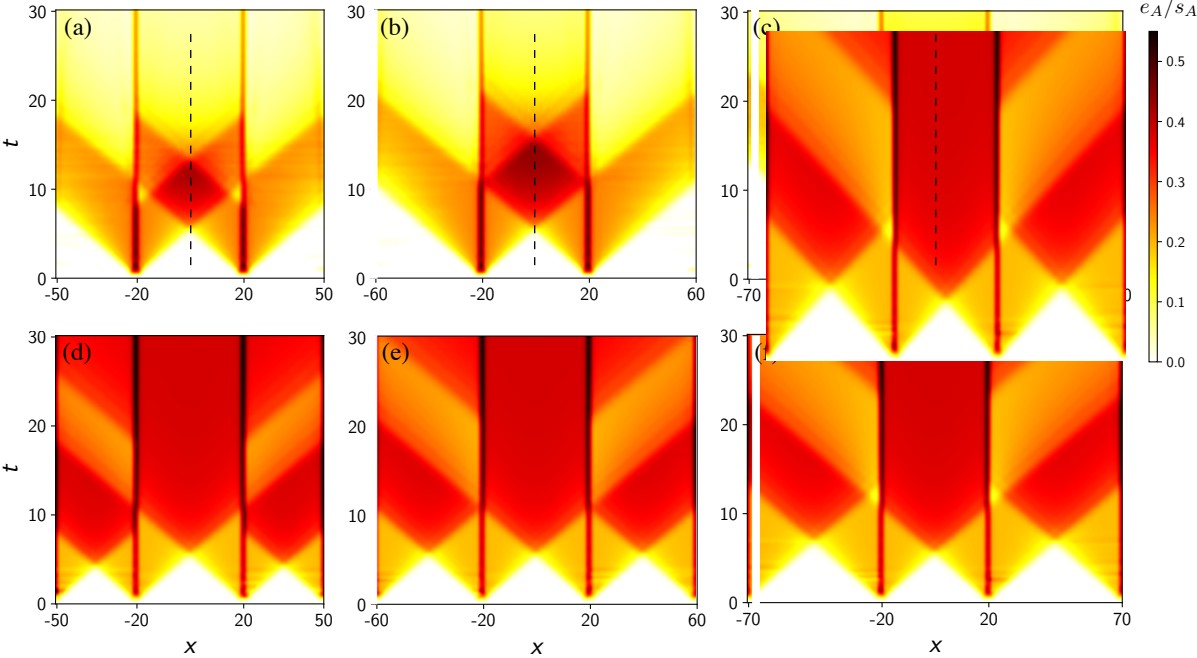

Figure 4: Time evolution of the negativity contour using Gaussian formula (first row) and the entanglement contour (second row) for various sizes of subsystem $B$. We use a total system size to be 1200 sites, $L_A = 40$, and $L_B = 60, 80, 100$ (from left to right, respectively) in the configuration shown in Fig. 3. Finite size effects cause recurrences in the negativity contour, so we only show times smaller than the total system size.

where $f^{R/L}$ are the two fermion species that live on each site and anti-periodic boundary condition $f_{i+L}^{R/L} = -f_i^{R/L}$ is always imposed. We use the trivial gapped state where $t_2 < t_1$ and quench into the gapless Hamiltonian by taking $t_2 = t_1$. Since the initial state is trivial, the two subsystems are initially unentangled. We compare the negativity and entanglement contours in Fig. 4. It is evident from the negativity contour (first row of Fig. 4) that the quantum correlation between $A$ and $B$ is transient. However, the entanglement contour (second row of Fig. 4) saturates to a constant (thermal) profile. This phenomenon can be also seen by comparing the logarithmic negativity and the entanglement entropy which are plotted in Fig. 5(b). This manifestly demonstrates that the system $A \cup B$ is in a classical (zero-entanglement) thermal state. In what follows, we discuss the quasi-particle picture as an effective description for the dynamics of quantum entanglement in integrable models including the free fermion systems.

In the spacetime scaling limit, the dynamics of entanglement in integrable systems are described by the quasi-particle picture [113–116]. In this effective description, entanglement is carried by local free-streaming quasi-particles. When a quasi-particle is contained in a region, the entanglement grows accordingly if its "pair particle" is outside of the region. This has been

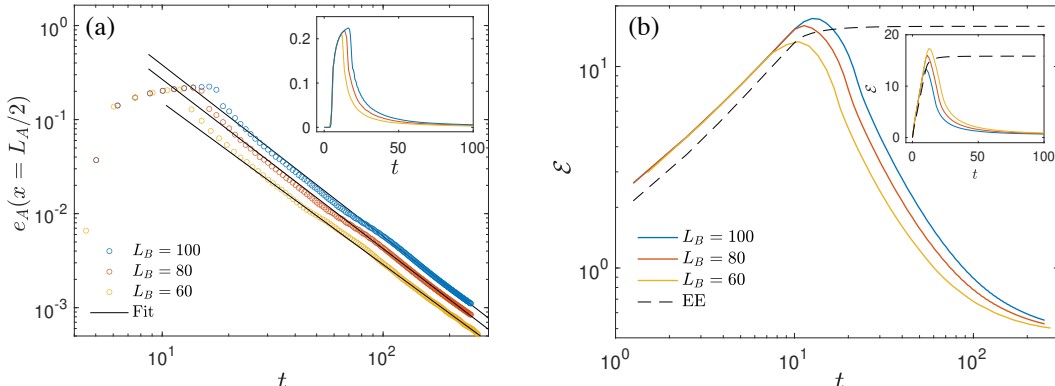

Figure 5: (a) The negativity contour at the center of subsystem $A$ (in Fig. 3) as a function of time for various sizes of subsystem $B$. This plot can be thought of as a single slice of Fig. 4 (see the dashed lines in the upper row of the same figure). The power-law fit (black lines) is given by $e_A \sim t^{-\alpha}$ where $\alpha = 1.8 \pm 0.1$. (b) The logarithmic negativity as a function of time. As a reference, the von Neumann entanglement entropy (dashed line) is also plotted. In both panels, the scaling of axes of the main plots is logarithmic, while the axes of the insets are linear. Here, $L_A = 40$ and total system size is 1200.

made quantifiable for example in a single interval of length $l$ as [115]

$$S(t) = \sum_n \left[ 2t \int_{2|v_n|t<l} d\lambda\, v_n(\lambda) s_n(\lambda) + l \int_{2|v_n|t>l} d\lambda\, s_n(\lambda) \right], \tag{44}$$

where the sum is over quasi-particle species, $v_n(\lambda)$ is the velocity of the quasi-particle, and $s_n(\lambda)$ is a function dependent on the production rate of quasi-particles. The quasi-particle picture naturally suggests an entanglement contour that was originally considered in Ref. [117]. This is because the quasi-particles are additive in the von Neumann entropy, so the contour is simply the contribution of the quasi-particles from the given subregion

$$s_A(x, t > 0) = \frac{1}{2} \sum_n \int d\lambda \Big( \Theta(2|v_n|t - x) + \Theta(2|v_n|t - l + x) \Big) s_n(\lambda),$$

where $A = (0, l)$ and $\Theta$ is the Heaviside step function. It is not hard to convince oneself that this quasi-particle definition of the contour is exactly the same as the derivative formula (7).

Interestingly, a quasi-particle picture has also been constructed for logarithmic negativity [118], so one may analogously find a negativity contour for quasi-particles corresponding to the the derivative formula (9). By construction, negativity contours based on the quasi-particle picture are bounded from below by zero. This means that (9) is a valid negativity contour for all integrable theories that follow the quasi-particle picture following a quench. For simplicity, we consider adjacent intervals $A_1 = (-l_1, 0)$ and $A_2 = (0, l_2)$ and find

$$e_{A_2}^{(f/b)}(x, t > 0) = \frac{1}{2} \sum_n \int d\lambda \left( \Theta(2|v_n|t - x) - \Theta(2|v_n|t - l_1 - x) \right) \varepsilon^{(f/b)}(\lambda), \tag{45}$$

where the superscript $(f/b)$ corresponds to the fermionic and bosonic negativities and $\varepsilon$ is the momentum-dependent contribution of quasi-particles to the logarithmic negativity. This solution corresponds to a moving "box" of quasi-particles. In general, the box will spread in space over time because of the momentum dependence of $\varepsilon$. We find (45) to be justified in

Fig. 4 where we have numerically computed the negativity contour for free fermions after a global quench. The decaying behavior of the negativity contour as a function of time is plotted in Fig.5(a), where we realize that the negativity contour does not relax all the way back to zero and its long time behavior at the center of subsystem $A$ is described by a power-law as in $e_A \sim t^{-\alpha}$, $\alpha = 1.8 \pm 0.1$. We observe the same power-law behavior with a similar exponent for other points inside subsystem $A$ or $B$ as long as we are away from the entanglement cuts. This can be attributed to the nonlinear dispersion relation that dictates some quasi-particles to move faster than others. We derive a $t^{-2}$ power law decay of the negativity contour for a related free fermion quench in Appendix C. We should note, however, that the overall logarithmic negativity (as shown in Fig. 5(b)) does not show a power-law behavior, because the main contribution comes from the $e_A(x)$ near the entanglement boundaries which do not obey a power-law. Another noteworthy difference between negativity and entanglement contours in Fig. 4 is that the negativity contour does not have the two extra light cones at the leftmost and rightmost ends of the subsystem $B$ which are present in the entanglement contour. This behavior of the negativity contour is consistent with the fact that the negativity is a measure of mutual correlation between $A$ and $B$ and indeed nothing should emerge at the interface between $B$ and $C$ (c.f. Fig. 3).

There is also significant interest in understanding entanglement dynamics in non-integrable theories where the quasi-particle picture fails. In these systems, entanglement is not carried by coherent localized objects. Rather, there are hints from holographic conformal field theories [11, 119] and random unitary circuits [120–122], that the dynamics of entanglement in ergodic quantum systems are carried by non-local membrane-like objects. In the former, the entanglement is captured by the bulk extremal surface, while in the latter, entanglement is captured by the "line-tension" of a membrane in spacetime. It would be fascinating to understand the entanglement and negativity contours in these systems. We leave more thorough analysis to future work.

## 6 Discussion

In this work, we have laid the foundation for studying the entanglement structure of mixed states at a quasi-local level through the introduction of the negativity contour. We have done so by constructing a Gaussian formula for free fermions analogous to the original work of Ref. [61] and then generalizing to generic many-body systems with a "derivative formula." We have shown that this derivative formula for the negativity contour along with its analog for the entanglement contour accurately reproduce the results of the Gaussian formulas. We note that even though the derivative formulas are more general, it is significantly more computationally efficient to use the Gaussian formulas when working in free theories.

We have applied our formalism to various quantum systems. In Fermi surface systems, we have seen how the negativity contour may shed light on the quantum-to-classical crossover [88]. The inverse temperature $\beta$ is manifestly the length scale at which quantum correlations disappear. There appear to be violations of this in non-Fermi liquids with a dynamical critical exponent. It would be fascinating to investigate this further, perhaps variationally using candidate wave functions [123, 124]. Finally, we have studied the dynamics of the negativity contour following global quantum quenches in integrable systems. While the dynamics of free fermions are well understood, this serves as a proof of principle for more novel non-integrable systems, and exciting prospects for future work.

An apparent disadvantage of the entanglement contour is that it is agnostic to multipartite entanglement. This should play a larger role in studies of strongly-interacting systems, though it is presently unclear how appropriate it is to apply a local entanglement density function to

systems that contain multipartite entanglement.

## Acknowledgments

The authors would like to acknowledge insightful discussions with Jesse Cresswell, Tarun Grover, Ali Mollabashi, Max Metlitski, Paola Ruggiero, T. Senthil, Erik Tonni, and Ashvin Vishwanath. JKF and SR thank the Yukawa Institute for Theoretical Physics (YITP) for hospitality during the completion of this work.

**Funding information**   This work was supported in part by the National Science Foundation under Grant No. DMR-1455296, and under Grant No. NSF PHY-1748958. SR is supported by a Simons Investigator Grant from the Simons Foundation. HS is supported through a Simons Investigator Award to Ashvin Vishwanath and Senthil Todadri from the Simons Foundation.

## A   Negativity contour of Gaussian states

In this appendix, we present a scheme to calculate the negativity contour for fermionic Gaussian states (free fermions). The following procedure works efficiently for any quadratic Hamiltonian of the form $\hat{H} = \sum_{i,j} t_{ij} \psi_i^\dagger \psi_j + \text{H.c.}$ [125]. The only input needed to calculate the negativity contour in this case is the correlation matrix. We first explain how to construct the correlation matrix for thermal equilibrium states and out-of-equilibrium states as a result of a global quantum quench. Next, we discuss how the negativity contour of a given correlation matrix can be computed.

The reduced density matrix $\rho$ of a Gaussian state is fully characterized by the single particle correlation matrix [126],

$$C_{rr'}(t) = \langle \psi_r^\dagger \psi_{r'} \rangle = \text{tr}(\rho \psi_r^\dagger \psi_{r'}). \tag{A46}$$

For a thermal state, we have

$$C_{rr'} = \sum_n f(\epsilon_n) \, u_n^*(r) \cdot u_n(r'), \tag{A47}$$

where $|u_n\rangle$ are single particle eigenstates $\hat{H} |u_n\rangle = \epsilon_n |u_n\rangle$, $u_n(j) = \langle j|u_n\rangle$ is the value of the wave function at site $j$, and $f(x) = (1 + \exp(x/T))^{-1}$ is the Fermi-Dirac distribution function. For the ground state, the Fermi-Dirac distribution at zero temperature automatically enforces the summation to be over the negative energy states.

In the case of quench dynamics, the initial state is an eigenstate of Hamiltonian $\hat{H}_0$ and the subsequent unitary dynamics is governed by Hamiltonian $\hat{H}_1$. The Hamiltonians are quadratic which can be written in a diagonalized form as

$$\hat{H}_0 = \sum_{n=1}^N \tilde{\epsilon}_n g_n^\dagger g_n, \qquad \hat{H}_1 = \sum_{n=1}^N \epsilon_n f_n^\dagger f_n. \tag{A48}$$

We should note that these operators are related to real-space operators $\psi_i, \psi_i^\dagger$ via single-particle unitary transformations

$$g_n^\dagger = \sum_r \varphi_n(r) \psi_r^\dagger, \qquad f_n^\dagger = \sum_r \phi_n(r) \psi_r^\dagger, \tag{A49}$$

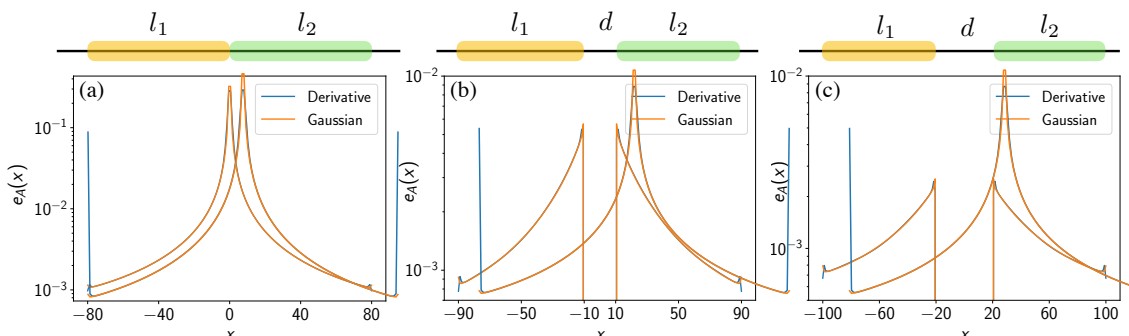

Figure 6: Negativity contour using Gaussian (A61) and derivative formulae (10) for two (a) adjacent and (b,c) disjoint intervals. We use a total system size of 400 sites, $l_1 = l_2 = 80$, and $d = 0, 20, 40$ (from left to right, respectively) in the configuration shown above each panel.

and

$$f_n^\dagger = \sum_m W_{nm} g_m^\dagger, \tag{A50}$$

where $W_{nm} = \sum_r \varphi_m^*(r)\phi_n(r)$. We compute the correlation matrix at a given time $t$

$$C_{rr'}(t) = \langle \psi_r^\dagger(t)\psi_{r'}(t)\rangle \tag{A51}$$

by finding the Heisenberg evolution of operators

$$\psi_r^\dagger(t) = e^{iH_1 t}\psi_r^\dagger e^{-iH_1 t} = \sum_n e^{i\epsilon_n t}\phi_n^*(r)f_n^\dagger. \tag{A52}$$

Suppose the initial state is $|\Psi_0\rangle = \prod_{n\in\text{occ.}} g_n^\dagger |0\rangle$, where occ. refers to the set of occupied states. Hence, we can write

$$C_{rr'}(t) = \sum_{n,n'} e^{i(\epsilon_n - \epsilon_{n'})t}\phi_n^*(r)\phi_{n'}(r')\langle f_n^\dagger f_{n'}\rangle \tag{A53}$$

$$= \sum_{n,n'} e^{i(\epsilon_n - \epsilon_{n'})t}\phi_n^*(r)\phi_{n'}(r') \sum_{m\in\text{occ.}} W_{nm}W_{n'm}^*. \tag{A54}$$

In the last line, we use the identity $\langle f_n^\dagger f_{n'}\rangle = \sum_{m\in\text{occ.}} W_{nm}W_{n'm}^*$.

Now that we have the correlation matrix, we explain our method of computing the negativity contour. We define the covariance matrix $\Gamma = \mathbb{I} - 2C$ which takes a block matrix form

$$\Gamma = \begin{pmatrix} \Gamma^{11} & \Gamma^{12} \\ \Gamma^{21} & \Gamma^{22} \end{pmatrix}, \tag{A55}$$

for a bipartite system $A = A_1 \cup A_2$ with $N = N_1 + N_2$ sites. Here, $\Gamma^{11}$ and $\Gamma^{22}$ denote the reduced covariance matrices of subsystems $A_1$ and $A_2$, respectively; whereas $\Gamma^{12}$ and $\Gamma^{21}$ contain the expectation values of mixed quadratic terms. We define the transformed covariance matrix as

$$\Gamma_\pm = \begin{pmatrix} -\Gamma^{11} & \pm i\Gamma^{12} \\ \pm i\Gamma^{21} & \Gamma^{22} \end{pmatrix}, \tag{A56}$$

corresponding to the partial transpose of the density matrix with respect to $A_1$. Using algebra of a product of Gaussian operators [109, 127], the new single particle correlation function associated with the normalized composite density operator $\Xi = \rho^{T_1}\rho^{T_1\dagger}/\mathcal{Z}_\Xi$ can be found by

$$C_\Xi = \frac{1}{2}\left[\mathbb{I} - (\mathbb{I} + \Gamma_+\Gamma_-)^{-1}(\Gamma_+ + \Gamma_-)\right], \tag{A57}$$

where the normalization factor is $\mathcal{Z}_\Xi = \text{tr}(\Xi) = \text{tr}(\rho^2)$. The logarithmic negativity is found by

$$\mathcal{E} = \ln\left[\mathcal{Z}_\Xi^{1/2}\text{tr}(\Xi^{1/2})\right] = \ln\text{tr}(\Xi^{1/2}) + \frac{1}{2}\ln\text{tr}(\rho^2). \tag{A58}$$

In terms of eigenvalues of the correlation matrices, we can write

$$\mathcal{E} = \sum_{j=1}^{N}[R(\xi_j; 1/2) + \frac{1}{2}R(\zeta_j; 2)], \tag{A59}$$

where

$$R(\lambda; q) = \ln\left[\lambda^q + (1-\lambda)^q\right], \tag{A60}$$

is a function with the property that $R(\lambda; q) \geq 0$, $q \leq 1$ and $R(\lambda; q) \leq 0$, $q \geq 1$. Here, $\zeta_j$ and $\xi_j$ are eigenvalues of the original correlation matrix $C$ in Eq. (A46) and the transformed matrix $C_\Xi$ (A57), respectively. We can use the eigenstates of $C$ and $C_\Xi$ to find a spatial decomposition of the negativity, i.e., an ansatz for the negativity contour which potentially satisfies the five criteria discussed in Sec. 2. Let $U_j(r)$ and $V_j(r)$ be eigenstates of $C_\Xi$ and $C$ with eigenvalues $\xi_j$ and $\zeta_j$, respectively. Given these facts, we introduce the following quantity

$$e_A(r) = \sum_{j=1}^{n}[|U_j(r)|^2R(\xi_j; 1/2) + \frac{1}{2}|V_j(r)|^2R(\zeta_j; 2)], \tag{A61}$$

as a candidate formula for the negativity contour of fermionic Gaussian states. We should note that $R(\lambda; 1/2) \geq 0$, while $R(\lambda; 2) \leq 0$ and our ansatz is not guaranteed to be positive. However, we always observe that this expression is positive. Moreover, the completeness of $\{U_j(r)\}$ and $\{V_j(r)\}$ eigenvectors implies $\sum_{r=1}^{N} e_A(r) = \mathcal{E}$. The invariance under local unitary operators is also guaranteed by the orthogonality of eigenvectors. Figure 6 shows a comparison of the derivative formula (9) and the Gaussian expression for the negativity contour of two intervals on a free fermion chain. The two quantities agree quite well away from the boundaries for both adjacent and disjoint intervals. It is interesting to note that within each interval the contour is monotonically increasing as we move towards the other interval.

We finish this appendix by providing analytic expressions for the negativity contour of tripartite geometry in the ground state of free fermions. For two adjacent intervals of lengths $l_1$ and $l_2$, the contour is found to be

$$e_A(x) = \frac{l_2/4}{(l_1/2 - x)(l_2 + l_1/2 - x)}, \tag{A62}$$

where $|x| \leq l_1/2$ defined over the interval $A$. It is easy to check that the above quantity satisfies

$$\int_{-l_1/2}^{l_1/2} e_A(x)dx = \frac{1}{4}\ln\left(\frac{l_1 l_2}{l_1 + l_2}\right), \tag{A63}$$

which is the familiar expression for a tripartite negativity of two adjacent intervals in a CFT with $c = 1$. Moreover, the negativity contour of two disjoint intervals (separated by distance $d$) is given by

$$e_A(x) = \frac{l_2/4}{(d + l_1/2 - x)(d + l_2 + l_1/2 - x)}, \tag{A64}$$

which reproduces the expected value for the logarithmic negativity [128],

$$\int_{-l_1/2}^{l_1/2} e_A(x)dx = \frac{1}{4}\ln\left(\frac{(l_1+d)(l_2+d)}{(l_1+l_2+d)d}\right). \tag{A65}$$

We should note that in contrast to the case of two adjacent intervals (A62) the negativity contour of two disjoint intervals does not diverge anywhere within its domain $|x| \leq l_1/2$ and it is bounded from above where the maximum value is $e_A = l_2/4d(l_2+d)$ at the right end, i.e., $x = l_1/2$.

## B  Universality of entanglement contours

In the main text, we mainly discussed the negativity contour. Here, we would like to compare the derivative formula as an entanglement contour (which satisfies all requirements for a contour [70]) with free fermion and boson Gaussian formulas developed in Refs. [3,63]. To this end, we consider a reduced correlation matrix $C_{ij} = \langle f_i^\dagger f_j\rangle$ of a subregion of length $l$ on an infinite free fermion chain. The Gaussian formula is given by

$$s_A(x) = \sum_{j=1}^{l} |U_j(x)|^2 s(\nu_j), \tag{B66}$$

where $\{U_i(j), 0 \leq \nu_i \leq 1\}$ is a set of eigenvectors/eigenvalues of the correlation matrix $C_{ij}$ and $s(x) = -x\ln x - (1-x)\ln(1-x)$. Similarly, for a bosonic chain we take reduced correlation matrices $Q_{ij} = \langle q_i q_j\rangle$ and $P_{ij} = \langle p_i p_j\rangle$ of a subregion of length $l$ on an infinite Harmonic chain [62,63] where $q_i$ and $p_i$ denote the canonical variables on each site such that $[q_m, p_n] = i\delta_{mn}$. The Gaussian formula for bosonic entanglement contour is given by

$$s_A(x) = \sum_{j=1}^{l} |V_j(x)|^2 \tilde{s}(\sigma_j), \tag{B67}$$

where $\{V_i(j), \sigma_i \geq 1/2\}$ is a set of eigenvectors/eigenvalues of the matrix $\Xi Q\Xi$ or $\Pi P\Pi$ (which are simultaneously diagonalizable) where

$$\Xi^2 = Q^{-1/2}(Q^{1/2}PQ^{1/2})^{1/2}Q^{-1/2}, \qquad \Pi^2 = P^{-1/2}(P^{1/2}QP^{1/2})^{1/2}P^{-1/2}, \tag{B68}$$

and $\tilde{s}(x) = (x+1/2)\ln(x+1/2) - (x-1/2)\ln(x-1/2)$. Notice that the entanglement entropy is $S = \sum_j \tilde{s}(\sigma_j)$.

The numerical results are plotted in Fig. 7. In the case of free fermions, we impose an anti-periodic boundary condition. In the case of Harmonic chain, we impose a periodic boundary condition while adding a small mass $\omega$ to avoid diverging correlators (as is usually done in the literature [62,63]). We observe that the derivative and Gaussian formulas match very well for the free fermion chain, while they differ quite a bit for the harmonic chain. Also, as we see in the right panel of Fig. 7, the universal form of the entanglement contour in CFT,

$$s_A(x) = \frac{c}{3}\frac{l/2}{l^2/4 - x^2} \tag{B69}$$

(shown as a solid black line) gives a good fit to the entanglement contour of the fermion or boson lattice. We believe that the slight deviation of the derivative formula for the harmonic chain from the CFT expression is due to the presence of the mass term.

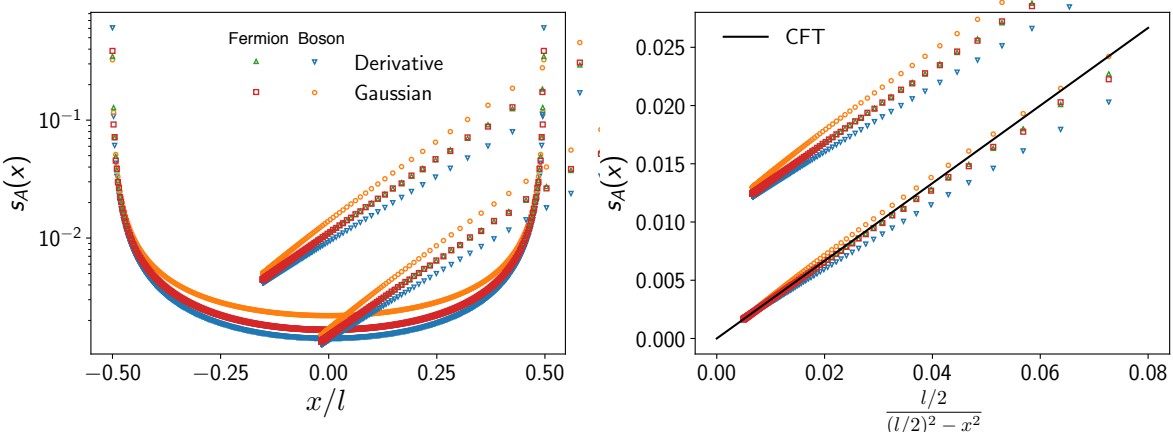

Figure 7: Entanglement contour using Gaussian and derivative formulae (7) for a finite region of length $l = 400$ in infinite bosonic and fermionic chains. The bosonic calculation is done for a chain of coupled harmonic oscillators (the so-called Harmonic chain [63]) and the fermionic calculation is carried out for a free fermion chain. The solid black line in the right panel is the continuum limit expression for the contour in CFT (B69) with $c = 1$. For the Harmonic chain, we added a mass term $\omega = 4 \times 10^{-4}/l$ to avoid singular behavior in the correlator $\langle q_i q_j \rangle$.

Let us now discuss the mysterious confluence of entanglement contour functions in holographic field theories. There are three proposals for the entanglement contour to compare, the derivative formula generalized to spheres in all dimensions [69, 70], the formula from bulk modular flow [66, 69], and the formula derived from explicit realizations of bit threads configurations [70]. We only consider ball-shaped regions where all of these formulas are well-defined. Furthermore, we only apply the derivative formula in $1 + 1d$.

The bulk modular flow construction states that entanglement contour can be computed by identifying points in the boundary interval with points on the Ryu-Takayanagi surface. The contour function is computed by the area of the subregion of the Ryu-Takayanagi surface associated to the boundary point. Finally, the bit thread construction involves constructing an explicit bit thread configuration. Bit threads are an alternative formulation of holographic entanglement entropy in which the entanglement is computed by a maximization over bit thread configurations [129]

$$S_A = \max_v \int_A v, \tag{B70}$$

where $v$ is a divergenceless vector field in the bulk with its norm bounded above by $(4G_N)^{-1}$. This presents a natural (though usually computationally intractable) contour function in holography

$$s_A(x) = |v(x)|. \tag{B71}$$

The authors of Ref. [70] used the geodesic flow construction of Ref. [130] which had bit threads configured so they follow bulk geodesics connecting the asymptotic boundary to the Ryu-Takayanagi surface. Remarkably, regardless of the construction, one finds the entanglement contour to be

$$s_A(r) = \frac{c}{6} \left( \frac{2R}{R^2 - r^2} \right)^d. \tag{B72}$$

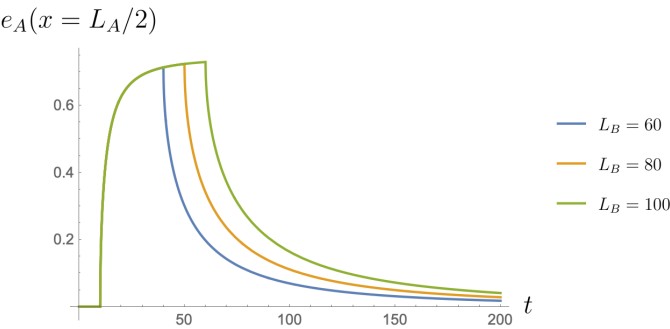

Figure 8: The negativity contour for the free fermion Hamiltonian (C74). We take $L_A = 40$ to provide the same parameters as Fig. 5 which has a very similar form. The late-time decay is proportional to $t^{-2}$. Note that the time scales for this model are twice that of the SSH quench due to the factor of two difference in the low energy dispersion relations of the models.

## C  Decay of negativity contour after global quench

In this appendix, we compute the decay of the negativity contour for a free fermionic system where we know the precise occupation number density, $\rho(k)$, after the quench. This allows us to compute the entanglement content because for free systems [118]

$$\varepsilon^{(f/b)}(k) = \pm \log\left(\pm \rho(k)^{1/2} \mp (1-\rho(k))^{1/2}\right), \tag{C73}$$

where the $\pm$ are for fermionic and bosonic negativity respectively. We consider the following free fermion Hamiltonian

$$H = -\frac{1}{2}\sum_i \left[\left(c_i^\dagger - c_i\right)\left(c_{i+1}^\dagger + c_{i+1}\right) + h c_i^\dagger c_i\right]. \tag{C74}$$

The dispersion relation is $e(k) = \left(h^2 - 2h\cos k + 1\right)^{1/2}$, so the quasi-particle velocities are

$$v(k) = \frac{\partial e(k)}{\partial k} = \frac{h\sin(k)}{\sqrt{h^2 - 2h\cos(k) + 1}}. \tag{C75}$$

We quench from the infinite gap ($h \to \infty$) product state to the critical point ($h = 1$). The occupation number density from the generalized Gibbs ensemble for this quench is [131–133]

$$\rho(k) = \frac{1}{2}\left(\frac{\cos(k) - 1}{\sqrt{2 - 2\cos(k)}} + 1\right). \tag{C76}$$

Then, using the general formula for the negativity contour for the quasi-particle picture (45), we find that the contour decays as $t^{-2}$ at late times, very close to the power law decay numerically observed for the quench in Sec. 5. The contour at all times for a range of parameters is shown in Fig. 8.

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
