# Peer review of "The negativity contour: a quasi-local measure of entanglement for mixed states"

_SciPost Physics, doi:SciPost Phys. 8, 063 (2020)_

## Round 1 · Referee Report · Anonymous (Referee 1) · 2020-2-5

Report

In this manuscript, Kudler-Flam and company investigate the negativity contour for several models, including ones which have a Fermi surface. The negativity contour can be considered a generalization of the entanglement contour, which has been used to investigate which degrees of freedom contribute to the entanglement entropy. This generalization is particularly useful given the rise of interest in mixed states and thermalization of quantum many-body systems. I am confident this work is solid and timely and will be of interest to the community. I recommend publication after a few minor issues, listed below, have been addressed.

Requested changes

1) The authors should add a sentence about negativity and advantages of using logarithmic negativity over it.

2) The authors should include a Figure for Eq. 14.

3) The authors should consider citing Phys. Rev. B 100, 241108(R) 2019, which investigates entanglement contours after a quench in momentum-space instead of real-space and show only momentum-space degrees of freedom near the Fermi surface contribute.

4) How does logarithmic negativity decay with time (for fixed position in Fig. 4)? It would be nice for the authors to another plot at fixed position as a function of time.

5) f and f^dagger do not appear to be defined near eq. 42. The authors should define these operators.

---

## Round 2 · Referee Report · Anonymous (Referee 1) · 2020-3-11

Report

The authors have addressed all my previous comments. At this time, I believe the manuscript is acceptable for publication.

---

## Round 2 · Author Response

We thank the referee for their valuable feedback. Below, we have addressed all minor issues
raised.

---

## Round 2 · List of Changes

1) We have added a footnote explaining our motivation for studying logarithmic negativity over entanglement negativity on page 3. 2) We note that equation (14) has already appeared in Figure 2. We have made this more clear in the caption of Figure 2 by adding a reference to the equation. 3) We have added a footnote discussing and citing this reference on page 11. 4) We have added analysis of the decay of the negativity contour in time in Section V. We have added the plot (Figure 5) suggested by the referee. Furthermore, we have added Appendix C, in which we use our analytic formula for the negativity contour to derive a $t^{-2}$ power law decay of the negativity contour for a related quantum quench. 5) We have added definitions of these operators below equation (42).

---

## Editorial Decision

published